# Effects of Biochar on Soil Inorganic Phosphorus Components, Available Phosphorus, Enzyme Activities Related to Phosphorus Cycle, Microbial Functional Genes, and Seedling Growth of *Populus euphratica* under Different Water Conditions

Yuxian Fan [1], Yudong Chen [1] and Guanghui Lv [1,2,3,*]

1. College of Ecology and Environment, Xinjiang University, Urumqi 830046, China; 18016844684@163.com (Y.F.); cyd666@stu.xju.edu.cn (Y.C.)
2. Key Laboratory of Oasis Ecology of Education Ministry, Xinjiang University, Urumqi 830017, China
3. Xinjiang Jinghe Observation and Research Station of Temperate Desert Ecosystem, Ministry of Education, Jinghe 833300, China
* Correspondence: guanghui_xju@sina.com

**Abstract:** Cow dung is a kind of high quality and renewable biological resource. Biochar made from cow dung can be used as a soil amendment to improve soil nutrient status. The relationship between soil water and phosphorus is very close, and the water status determines the form, content, and availability of phosphorus. In order to investigate the effects of biochar on soil inorganic phosphorus components, available phosphorus, enzyme activities related to the phosphorus cycle, microbial functional genes, and seedling growth under different soil water conditions were investigated. Field experiments were carried out by setting different water conditions (30%, 60%, and 100%) and biochar addition (0 t hm$^{-2}$, 2.63 t hm$^{-2}$, 5.26 t hm$^{-2}$, and 7.89 t hm$^{-2}$). The results showed that applying biochar significantly increased the soil's accessible phosphorus content and the phosphorus content in both the aboveground and subsurface parts of *P. euphratica* seedlings. This is mainly attributable to biochar's direct and indirect effects on soil properties. Because biochar is naturally alkaline, it raises soil pH and reduces acid phosphatase activity in the soil around *P. euphratica* seedlings in the rhizosphere. Perhaps the alkaline phosphatase level first showed an upward trend due to the combined impacts of water and biochar, and then it started to decline when the biochar addition was increased. Soil phosphorus functional genes *phoC*, *phoD*, *gcd*, and *pqqc* had an increase in copy number with biochar addition but not without treatment. Indirectly, the biochar treatment increased the soil's phosphorus availability by increasing the population of the phosphate-solubilizing bacteria *Fusarium* and *Sphingomonas*. Soil phosphorus availability is positively affected by biochar under various water conditions. This impact is due to chemical and microbiological mechanisms.

**Keywords:** biochar; P effectiveness; phosphatase; microbial community structure; P functional gene

## 1. Introduction

*Populus euphratica* is a deciduous broad-leaved tree species widely distributed in Xinjiang desert. It is the result of the survival struggle under the arid desert climate and poor soil conditions, thus forming its ecological characteristics of liking light and resisting heat, enduring certain low temperatures, resisting atmospheric drought, resisting wind and sand, resisting salt and alkali, and adapting to desert conditions [1]. A precious forest resource in Xinjiang's dry desert region, *P. euphratica* is pivotal to the region's ongoing ecological transformation, adaptation, and population establishment of *P. euphratica* in harsh habitats [2]. *P. euphratica* seedlings have a number of challenges in their artificial planting environment, including a lack of soil nutrients and a poor ability to retain fertilizer, as well as dry climate and sandy soil features. In addition, planting in the same spot year after year depletes soil

nutrients, particularly phosphorus, which in turn stunts the regeneration and development of *P. euphratica* populations, which in turn impedes ecological restoration efforts in dry desert regions. Soil phosphorus deficit can be addressed by applying phosphate fertilizer, but this practice is costly, leads to water body eutrophication, and increases greenhouse gas emissions; therefore, it is not sustainable in the long run [3,4].

One of the most important minerals for crops is phosphorus. *P. euphratica* seedlings can benefit from phosphorus in a number of ways, including faster growth, improved root development, increased root absorption capacity, and, to a certain extent, increased stress tolerance [5]. The majority of plant-available phosphorus comes from soil, which contains only a trace amount of the element (0.02%~0.2%, or $P_2O_5$ 0.05%~0.46%). The forest soil was amended with phosphate fertilizer to promote seedling growth and increase the benefits of afforestation. Chemical phosphate fertilizers tend to concentrate in the soil as insoluble compounds due to phosphorus's ease of fixation, leading to a low rate of phosphate fertilizer usage during the growing season—typically between 10% and 25% [6].

Cow dung is a kind of high-quality renewable biological resource. The rational utilization of cow dung can make good use of it. Pollution of the ecological environment and inefficient use of resources are likely outcomes of careless disposal [7]. The application of cow dung to the soil also raises the danger of nitrogen and phosphorus pollution of surface water and groundwater due to the high concentration of soluble forms of these elements in the dung [8]. Greenhouse gas emissions from composting, including carbon dioxide and ammonia, will further worsen the trend of global warming [9].

As a new substance, biochar has found numerous applications in the agricultural and environmental sectors in recent years. The pyrolysis of diverse biomass (e.g., agricultural straw, animal dung, wood, etc.) between 300 and 1000 °C produces biochar, a porous, refractory substance rich in organic carbon [10]. More and more studies have shown that cow dung carbonization technology can be widely used in soil improvement [11,12]. The use of sub-high-temperature anoxic retorting technology to prepare cow dung into biochar, and then its use for the preparation of carbon-based fertilizers, carbon-based soil amendments, and other products, can not only optimize the efficient use of cow dung resources but also use the special properties of biochar to help improve soil nutrient status and increase crop efficiency. Biochar made from cow dung contains more organic nitrogen, soluble phosphate, and potassium than biochar made from plants, and about 90% of phosphorus will be preserved during the pyrolysis of cow dung, which promotes the recycling of phosphorus as biochar after the pyrolysis of cow dung [13].

Because biochar is high in nutrients, it may be utilized directly as a source of nutrients for soil microorganism growth and development, hence increasing the population of soil microorganisms [14]. Indirectly, biochar can change the diversity of soil microbes. For instance, biochar's distinct structure enables it to have a large number of pores and a high specific surface area. These openings have the potential to alter the soil microbes' natural habitat by providing them with a new kind of housing and protection [15]. Numerous alterations have been seen in the organization of the soil microbial community following the application of biochar, according to earlier research [16–18]. Biochar can also improve the effectiveness of soil phosphorus by affecting the number of phosphate-solubilizing microorganisms, low molecular organic acid content, and enzyme activity. Different factors such as biochar type, biochar addition amount, addition duration, and soil type will have different degrees of influence [19,20]. Soil microbes that solubilize phosphate include actinomycetes, phosphate-solubilizing fungi (like *Penicillium*), and phosphate-solubilizing bacteria (like *Bacillus*) [21]. Soil phosphorus availability is influenced by phosphate-solubilizing bacteria, which make up a significant portion of microorganisms (ranging from 1% to 50%) [22]. The percentage of soil fungi that are phosphate-solubilizing ranges from 0.1% to 0.5%, and their capacity to solubilize phosphate is greater than that of phosphate-solubilizing bacteria [23]. Some microbes that dissolve phosphates may also mineralize organic phosphorus, whereas others can dissolve inorganic phosphorus compounds. The *gcd* and *pqq* genes, which include *pqqB* and *pqqC*, are responsible for the solubilization of insoluble inorganic

phosphorus in phosphate-solubilizing microbes. On the other hand, the *pho* genes, which encode multi-component phosphate transporters, are primarily responsible for phosphate uptake and transport. Common molecular markers include the *phoC*, *phoD*, *gcd*, and *pqqC* genes, which are functional and contribute to microbial phosphorus transformation [24,25].

Additionally, water is the key component that contributes to a decrease in agricultural output, particularly in semi-arid and dry regions. The location of the Xinjiang Uygur Autonomous Region may be found at the very center of Eurasia. The weather is characterized by drought conditions, since there is very little to no precipitation and there are limited water sources. When it comes to the development of Xinjiang, one of the most important issues is the improvement in agricultural soil water exploitation. This is because the shortage of water resources is a big element that is restricting the province's ability to expand in a sustainable manner. As a result of the fact that the capacity of plants to take in and move water has a considerable influence on the physiology and production of plants, the conservation of soil water is an important component of agricultural productivity. According to research, the application of biochar may have an effect on the soil's ability to retain water because of its high specific surface area, hydrophilic structure, and high porosity. The results of these research studies, on the other hand, are not consistent with one another. Following the application of biochar, a number of studies have recorded a variety of impacts on the soil's ability to retain water, including increases, declines, or no changes at all. As a result, it is essential to acquire further knowledge on the impact that biochar has on the properties of soil water. The beneficial effects of biochar on agricultural systems are becoming more well-known, while its influence on forest systems remains unclear. The majority of current biochar improvement research focuses on acidic soils; however, there is a dearth of literature on how biochar affects phosphorus availability in alkaline soils found in arid regions, namely gray desert soil. Using *P. euphratica* seedlings as a starting point, this study examined the impact of biochar on soil phosphorus availability, phosphatase activity, microbial community, and seedling growth under varied water conditions. The paucity of studies on the effects of biochar on phosphatase activity and phosphorus transformation has severely restricted the usage and promotion of biochar in alkaline soil regions in northern China, and the structure of soil microbes in alkaline desert soil (gray desert soil) under different water conditions. This is despite the extensive research on biochar's effects on soil microbes and phosphorus availability. Furthermore, the abundance of several functional genes in phosphorus-transforming microorganisms was determined and assessed using quantitative polymerase chain reaction (qPCR). The research conducted revealed the micro-ecological mechanism via which biochar alters the composition of the soil microbiome to enhance phosphorus availability at the roots of *P. euphratica* seedlings. Additionally, we gained insight into the potential micro-effects of biochar on *P. euphratica* seedlings in arid areas, which might be valuable for future seedling planting efforts.

We tested the hypothesis that, depending on the water conditions, biochar greatly increased the soil's available phosphorus. It is possible that biochar's impact on the soil's chemical and microbiological phosphorus processes is responsible for its capacity to increase soil phosphorus availability across a range of water conditions. Researchers used biochar to assess the soil's accessible phosphorus, inorganic phosphorus fractions, phosphatase activity, soil microbial community, and abundance of phosphorus transformation key genes in soil planted with 2-year-old *P. euphratica* seedlings. The experiment lasted a year and was designed to put these hypotheses to the test.

## 2. Materials and Methods

### 2.1. Study Area

The research area is located in a forest garden nursery in Midong District, Urumqi City, Xinjiang Uygur Autonomous Region, China (44°01′ N, 87°65′ E). The basic soil properties are as follows: pH: 8.01; bulk density: 1.23 g/cm$^3$; CEC: 19.48 cmol/kg; SOM: 20.25 g/kg;

total nitrogen: 1.28 g/kg; total phosphorus: 0.88 g/kg; total potassium: 18.39 g/kg; AN: 58.14 mg/kg; Olsen-P: 15.69 mg/kg; and AK: 336.21 mg/kg.

## 2.2. Biochar Materials

For their biochar experiment, researchers in the Xinjiang Uygur Autonomous Region collected raw cow dung from beef cattle farmers in Urumqi City's Midong District. The parameters of some of its traits are provided as follows: pH 7.8, specific surface area 3.08 m$^2$/g, pore size 2.07 nm, total nitrogen 4.24 g/kg, total phosphorus 3.58 g/kg, ash 16.48%, carbon 22.61%, hydrogen 5.54%, oxygen 26.65%, and nitrogen 1.36%. After a week of drying, the collected cow dung was burned anaerobically for two hours at 500 °C; then, it was held for another two hours before being cooled to room temperature. The resulting biochar was thoroughly mixed using a 2 mm sieve. The following variables were recorded for some of its traits: pH 10.14, specific surface area 7.36 m$^2$/g, pore size 3.47 nm, total nitrogen 10.68 g/kg, total phosphorus 6.45 g/kg, ash 22.69%, carbon 38.41%, hydrogen 4.26%, oxygen 8.06%, and nitrogen 3.88%.

## 2.3. Experimental Design

The experiment set three water levels, W30, W60, and W100, which were 30%, 60%, and 100% of the normal drip irrigation amount in the local field, and the normal drip irrigation amount in the local field was 3057 m$^3$ hm$^{-2}$. Under each water treatment, four biochar additions were set, BC0(CK), BC1, BC2, and BC3, and the biochar application rates were 0 t hm$^{-2}$, 2.63 t hm$^{-2}$ (35 percent carbon output from 7.5 t hm$^{-2}$ cow dung pyrolysis carbonization preparation, according to the present carbon manufacturing technique), 5.26 t hm$^{-2}$, and 7.89 t hm$^{-2}$. A randomized block design was used in this experiment. Each therapy was conducted three times. There was a 1 m distance between each plot, a 50 cm spacing between plants and rows, and a plot area of 7.5 × 10 m$^2$. Encircling the area were protective rows. The planting of 2-year-old *P. euphratica* seedlings was conducted in late March. The seedlings were chosen for their comparable plant height, full roots, and healthy roots. The cow dung biochar was artificially sprinkled on top of the soil before the *P. euphratica* seedlings were planted. Then, using a rotary tiller, it was thoroughly and uniformly blended into the soil. By adjusting the drip irrigation schedule and frequency in accordance with the local irrigation water management and the actual seedling growth of *P. euphratica*, we kept the plants from experiencing drought stress. Other than that, we followed the conventional cultivation management practices in our field. Seedlings of *P. euphratica* were gathered at the conclusion of their growth phase in late October, and their biomass (kg plant$^{-1}$) was determined by isolating their stems and roots. After 30 min in an oven set at 105 °C, the seedlings of *P. euphratica* were cooled to a consistent weight by drying them at 80 °C for 48 h. The sample was crushed into powder, digested and oxidized by HCLO$_4$-HNO$_3$, and colored by ammonium vanadate molybdate chromogenic solution. The color was compared at a 450 mm wavelength of an ultraviolet spectrophotometer (UV2802S, Shimadzu, Kyoto, Japan), so as to determine the phosphorus concentration in plant parts (leaves, stems, and roots) by colorimetry. It was necessary to collect soil samples from the root zones of each treatment. In each plot, a composite rhizosphere sample was taken after shaking free the loose dirt and brushing off any soil that was still adhering to the root. After passing through a 2 mm filter, the dirt was divided into two parts. One part was left to air-dry and was used for analyzing soil characteristics. The second part was preserved at a temperature of −80 °C for extracting soil DNA. The next step was to remove the stones and plant remnants.

## 2.4. Standard Soil Physical and Chemical Characteristics

A ratio of 1 part soil to 2.5 parts deionized water (*w/v*) was used to determine the pH of the soil using a Leici pH meter made in Shanghai, China. The total carbon was determined using combustion-infrared absorption spectroscopy. A mixture of potassium dichromate and sulfuric acid can be used to identify the organic components. The cation

exchange capacity was assessed at a pH of 7 using the ammonium acetate procedure. We used the Kjeldahl method to find the total nitrogen and the flame photometric method to find the total potassium. Both the alkaline hydrolysis diffusion technique and the neutral ammonium acetate solution extraction–flame photometer method were used in order to determine the amount of alkali-hydrolyzed nitrogen. The alkaline hydrolysis diffusion technique was utilized in order to determine the amount of accessible potassium. The quantities of aluminum and iron were calculated using inductively coupled plasma mass spectrometry.

### 2.5. Identifying Soil Phosphorus Properties

The total phosphorus content of the soil was ascertained through the use of phospho-molybdate colorimetry. We used a sodium bicarbonate extraction–molybdenum antimony colorimetric approach to find out how much accessible phosphorus was in the soil. Determination of inorganic phosphorus fractions in soil: Al-P was extracted with $1.0 \ mol \cdot L^{-1}$ $NH_4Cl$ solution; fe-P was extracted with $0.1 \ mol \cdot L^{-1}$ NaOH solution. Ca-P was extracted with $0.5 \ mol \cdot L^{-1}$ $H_2SO_4$ solution, and then the phosphorus concentration was determined by the Mo-Sb colorimetric method.

### 2.6. Determination of Soil Phosphorus Cycle-Related Enzyme Activity

The determination of soil acid phosphatase, neutral phosphatase, and alkaline phosphatase activity: Using the disodium phenyl phosphate colorimetric method, different buffers were used to prepare disodium phenyl phosphate, and the activity of acid phosphatase, neutral phosphatase, and alkaline phosphatase was determined at a wavelength of 660 nm.

Determination of phosphodiesterase activity: 1 mL of $CaCl_2$ solution and 5 mL of tris (hydroxymethyl) aminomethane-NaOH extractant were added to the culture medium with sodium bis-p-nitrophenyl phosphate solution and gently shaken up. After filtration, the absorbance of the culture medium was measured by a wind photometer at a wavelength of 410 nm.

Determination of pyrophosphatase activity: 5 mL 50 mol/L $Na_4P_2O_7 \cdot 10H_2O$ solution was added, then 5 mL buffer solution (pH = 8) and 25 mL 1 mol/L $H_2SO_4$ solution were added. The supernatant was centrifuged at 12000 rpm for 45 s, and 1 mL supernatant was taken to determine the concentration of phosphate.

Determination of phytase activity: 2 mL toluene was added, then 15 mL sodium hyaluronate solution was added, and then 50 mL $(NH_4)2SO_4$-$H_2SO_4$ solution was added. The shaker was shaken for 1 h, filtered, 10 mL of filtrate was taken out, moved into a 50 mL volumetric flask, 35 mL distilled water and 2 mL ammonium molybdate solution were added, the stannous chloride solution was added for the color reaction, and the wind-light photometer was used to determine the absorbance of the culture medium at a wavelength of 650 nm.

### 2.7. The Extraction of DNA, Amplification, Sequencing, and Handling of Sequence Data

The removal of phosphorus from the soil was accomplished by the use of a technique known as chloroform fumigation. Additionally, the phosphorus content of the soil microbial biomass was determined through the utilization of ammonium molybdate–ascorbic acid colorimetry.

Following the completion of the high-throughput sequencing, Beijing Baimaike Biotechnology Co., Ltd. (Beijing, China) manufactured the 16S rDNA PCR products. We utilized the TGuide S96 Magnetic Soil/Stool DNA Kit, which was made by Tiangen Biochemical Technology (Beijing) Co., Ltd. (Beijing, China), to collect soil DNA in a careful manner in line with the instructions provided by the manufacturer. The concentration and purity of the DNA were evaluated with the use of an ultraviolet photometer called the Nanodrop ND-2000, which was provided by Thermo Scientific of Wilmington, NC, which is located in the United States. Through the use of 1.8% agarose gel electrophoresis, the

amount and quality of the DNA extraction were taken into consideration. For the purpose of amplifying the hypervariable region V3-V4 of the 16S rRNA gene found in bacteria, primers were used. Both the 338F: 5′-ACTCCTACGGGAGGCAGCA-3′ and the 806R: 5′-GGACTACHVGGGTWTCTAAT-3′ primer sequences were used. Polymerase chain reaction (PCR) is often used by fungal communities in order to amplify certain portions of 18S rRNA that are exclusive to the internal transcribed spacer (ITS) region. This area is essential for the survival of fungi. The primer sequences are the ITS1-F: CTTGGTCATTTAGAGGAAGTAA and the ITS1-R: GCTGCGTTCTTCATCGATGC. During the polymerase chain reaction (PCR) procedure, the following components were utilized: 5-50 ng of DNA template, 0.3 µL of upstream primers (10 µM and 10 µM, respectively), 2 µL of 2 mmol L-1 dNTPs, 5 µL of KOD buffer, and 0.2 µL of KOD polymerase. The volume of the reaction was made up of a total of 10 µL. The procedure is repeated twenty times, with each cycle lasting five minutes at 95 degrees Celsius for pre-denaturation, thirty seconds for denaturation, thirty seconds for annealing, forty seconds for extension at 72 degrees Celsius, and seven minutes for final extension. Qseq-400 was used for quantification, and the Omega DNA purification kit (Ml-bio, Shanghai, China) was used for purification of the amplified products. On the Illumina Novaseq 6000 platform, the purified paired-end PCR products ($2 \times 250$ bp) were sequenced. The primer sequence was identified and removed using Cut Adapt. The UCHIME method was used to eradicate chimeras after the acquisition of PE measurements by USEARCH splicing. The aforementioned techniques generated high-quality readings, which we used for our next study. Using USEARCH (v10), we merged sequences with a similarity level higher than 97% into a single operational taxonomic unit (OUT), and we filtered all samples with OUTs lower than 2. When using classify-consensus-blast in QIIME2, the sequence may be annotated with at least 90% sequence similarity, 90% coverage, and 51% consistency. For the sequence categorization, we used the SILVA library (http://www.arb-silva.de, accessed on 5 December 2023). Raw sequence data reported in this paper have been deposited in the NCBI (accession number: PRJNA1098623).

### 2.8. Quantification of Functional Genes

The qPCR analysis was finished by Weikemeng Technology Group Co., Ltd., which is located in Shenzhen, China, as well. When we extracted soil DNA using a FastDNA SPIN kit(MP, CA, USA), we paid close attention to the directions provided by the manufacturer and followed them to the letter. For the purpose of determining the concentration and purity of DNA, a Nanodrop ND-2000 ultraviolet photometer was used under the microscope. The information on the primers that were used in the experiment is shown in Table S1. With the help of the AceQ® qPCR SYBR® Green Master Mix kit (Vazyme, Nanjing, China), the real-time fluorescence quantitative MA-6000 is able to duplicate three different gene amplification processes in order to efficiently extract DNA. In order to prepare the SYBR and primers, a mixture A technique was used. This approach included the addition of 10 µL of a $2 \times$ SYBR real-time PCR premixture, 0.4 µL of a PCR-specific primer F, and 0.4 µL of a PCR-specific primer R. Mixture A and a template sample that had been diluted, each with a volume of 8 µL, comprised the setup. Before annealing at 52 degrees Celsius, the real-time PCR reaction protocol consisted of five minutes of pre-denaturation at 95 degrees Celsius, fifteen seconds of pre-denaturation at 95 degrees Celsius, and thirty seconds of pre-denaturation at 60 degrees Celsius. Following the completion of each stage, data were collected. Following the conclusion of the test, we immediately conducted an analysis of the melting curve in order to verify that the reaction was specific. It was our responsibility to build plasmid standards and to establish standard curves. The values of the standard curve R that corresponded to *phoC*, *phoD*, *gcd*, and *pqqC* were 0.9974, 0.9982, 0.9984, and 0.992, respectively, according to the data.

### 2.9. Data Analysis

The data were analyzed statistically using SPSS 23.0, developed by SPSS, Inc. and located in Chicago, IL, USA. There was a one-way ANOVA as well as the LSD test. The

significance level was set at $p < 0.05$ *. To conduct the cluster analysis, the pheatmap package and the vegan R package were utilized, with the basis being the Spearman correlation matrix. Every gene associated with phosphorus in the bacterial and fungal phyla was subjected to Spearman correlation analysis. The R (4.3.0) software's igraph package was used to build the network, and Gephi(0.1.0)was used to visualize it.

## 3. Results

### 3.1. Soil Properties and Seedling Growth of P. euphratica

Tables 1–3 show that the addition of charcoal and water had a substantial impact on the soil's characteristics and the growth of *P. euphratica* seedlings. Under W60 water conditions, the CEC, SOM, total C, total Fe, AN, and AK were found to be 17%, 17%, 4%, 6%, 5%, and 4% higher than in the soil that did not have biochar added under W30 water conditions. The CEC, total carbon, total potassium, and total iron levels were all considerably elevated by 6%, 4%, 7%, and 6%, respectively, in the presence of W100 water. The root biomass, stem biomass, leaf biomass, and total biomass of *P. euphratica* seedlings were all considerably enhanced by 10%, 6%, 21%, and 9%, respectively, when grown in W60 water conditions as opposed to the soil devoid of biochar under W30 water conditions. The root biomass, stem biomass, leaf biomass, and total biomass of *P. euphratica* seedlings were significantly increased by 14%, 11%, 30%, and 15% under the W100 water condition. Compared with no biochar treatment, biochar treatment significantly improved soil properties and *P. euphratica* seedling growth. Compared with the soil without biochar addition, pH, CEC, SOM, total C, total N, total K, total Al, total Fe, AN, and AK increased by 0.3, 4%, 4%, 41%, 36%, 6%, 11%, 11%, 15%, and 17%, respectively, under BC2 biochar addition. The root, stem, leaf, and total biomass of *P.euphratica* seedlings increased significantly by 11%, 10%, 32%, and 15%, respectively. The root, stem, leaf, and total phosphorus contents of *P. euphratica* seedlings were significantly increased by 14%, 14%, 12%, and 13%, respectively.

**Table 1.** Soil properties affected by different biochar concentrations in different water environments.

| | pH | CEC (cmol/kg) | SOM (g/kg) | Total C (%) | Total N (g/kg) | Total K (g/kg) | Total Al (g/kg) | Total Fe (g/kg) | AN (mg/kg) | AK (mg/kg) |
|---|---|---|---|---|---|---|---|---|---|---|
| W30BC0 | 7.87 ± 0.02 g | 20.56± 0.12 f | 22.15 ± 0.15 f | 2.18 ± 0.02 k | 1.25 ± 0.01 h | 20.36 ± 0.05 f | 61.26 ± 0.11 f | 30.15± 0.15 f | 61.52± 1.56 h | 336.14 ± 5.36 i |
| W30BC1 | 7.95 ± 0.02 f | 21.63± 0.15 e | 23.63 ± 0.08 e | 2.36 ± 0.01 i | 1.59 ± 0.02 f | 21.45 ± 0.02 d | 62.66 ± 0.08 e | 33.45± 0.26 c | 68.15± 2.48 f | 358.48 ± 4.95 g |
| W30BC2 | 8.15 ± 0.01 b | 22.41± 0.17 d | 24.84 ± 0.13 d | 3.48± 0.02 g | 1.66 ± 0.01 d | 21.68 ± 0.06 c | 65.41 ± 0.07 c | 32.39± 0.18 d | 70.36± 2.84 e | 384.33 ± 5.12 f |
| W30BC3 | 8.11 ± 0.03 c | 21.35± 0.08 e | 22.14 ± 0.09 f | 4.01± 0.05 c | 1.83 ± 0.03 b | 22.15 ± 0.03 b | 65.15 ± 0.13 c | 31.82± 0.06 e | 75.96± 2.36 c | 394.52 ± 3.85 e |
| W60BC0 | 7.88 ± 0.02 g | 24.66± 0.06 b | 26.63 ± 0.11 b | 2.28 ± 0.03 j | 1.18 ± 0.01 j | 20.50 ± 0.04 f | 61.25 ± 0.25 f | 31.94± 0.14 e | 65.14± 1.95 g | 351.36 ± 3.45 h |
| W60BC1 | 7.95 ± 0.02 f | 24.17± 0.14 b | 26.48 ± 0.12 b | 2.69± 0.01 h | 1.63 ± 0.02 e | 20.36 ± 0.04 f | 63.48 ± 0.21 d | 34.69± 0.23 b | 71.51± 1.54 e | 383.69 ± 4.52 f |
| W60BC2 | 8.18 ± 0.01 a | 25.63± 0.18 a | 27.63 ± 0.07 a | 3.89± 0.02 d | 1.85 ± 0.03 a | 21.87 ± 0.02 c | 68.58 ± 0.09 a | 35.87± 0.16 a | 76.36± 1.69 c | 423.41 ± 5.63 b |
| W60BC3 | 8.05 ± 0.02 e | 23.45± 0.09 c | 25.15 ± 0.05 c | 4.59± 0.04 a | 1.75 ± 0.05 c | 21.36 ± 0.03 d | 67.14 ± 0.16 b | 35.14± 0.36 a | 83.14± 2.98 a | 488.39 ± 5.14 a |
| W100BC0 | 7.68 ± 0.01 i | 21.84± 0.11 e | 21.36 ± 0.16 g | 2.26± 0.05 j | 1.22 ± 0.02 i | 21.95 ± 0.05 c | 60.14 ± 0.14 g | 32.15± 0.26 d | 62.86± 2.24 h | 333.98 ± 2.89 i |
| W100BC1 | 7.86 ± 0.01 g | 21.54± 0.16 e | 25.36 ± 0.12 c | 2.46± 0.03 g | 1.43 ± 0.01 g | 22.47 ± 0.02 a | 63.55 ± 0.19 d | 31.96± 0.13 e | 65.15± 2.78 g | 406.66 ± 2.14 d |
| W100BC2 | 8.06 ± 0.01 e | 22.28± 0.15 d | 26.14 ± 0.07 b | 3.68± 0.01 e | 1.58 ± 0.02 f | 21.26 ± 0.03 e | 62.58 ± 0.13 e | 33.15± 0.11 c | 73.55± 1.15 d | 418.69 ± 4.36 c |
| W100BC3 | 8.09 ± 0.02 d | 20.63± 0.07 f | 25.69 ± 0.18 c | 4.25± 0.02 b | 1.66 ± 0.04 d | 20.14 ± 0.05 g | 63.14 ± 0.09 d | 32.48± 0.07 d | 79.63± 1.69 b | 386.93 ± 5.97 f |

Due to the small sample size (n = 3), all data are given as the mean ± standard error. Variations in the same column representing various treatments are indicative of a statistically significant difference ($p < 0.05$).

**Table 2.** Variations in *P. euphratica* seedling biomass (g dry weight plant$^{-1}$), as a function of water stress and biochar concentration in soils.

| | Root Biomass (g plant$^{-1}$) | Stem Biomass (g plant$^{-1}$) | Leaf Biomass (g plant$^{-1}$) | Total Biomass (g plant$^{-1}$) |
|---|---|---|---|---|
| W30BC0 | 115.51 ± 10.24 c | 268.24 ± 20.45 c | 63.68 ± 12.48 d | 447.43 ± 35.42 c |
| W30BC1 | 128.69 ± 11.62 b | 279.58 ± 18.69 b | 75.15 ± 10.87 c | 483.42 ± 33.69 c |
| W30BC2 | 133.48 ± 10.84 b | 297.63 ± 19.34 b | 89.58 ± 11.68 b | 520.69 ± 36.14 b |
| W30BC3 | 129.48 ± 12.59 b | 287.65 ± 20.87 b | 82.47 ± 14.41 c | 499.6 ± 37.56 b |
| W60BC0 | 127.69 ± 12.47 b | 284.24 ± 22.77 b | 80.36 ± 12.14 c | 492.29 ± 36.98 b |
| W60BC1 | 135.18 ± 10.78 a | 303.25 ± 21.68 a | 96.25 ± 10.35 b | 534.68 ± 38.15 b |
| W60BC2 | 143.65 ± 11.58 a | 316.69 ± 20.67 a | 118.48 ± 12.67 a | 578.82 ± 37.51 a |
| W60BC3 | 138.26 ± 12.96 a | 307.36 ± 19.89 a | 105.69 ± 15.47 a | 551.31 ± 39.69 a |
| W100BC0 | 133.68 ± 10.98 a | 300.14 ± 23.31 a | 90.47 ± 12.74 a | 524.29 ± 36.23 a |
| W100BC1 | 135.69 ± 11.06 a | 304.47 ± 19.86 a | 95.14 ± 11.01 a | 535.3 ± 37.05 a |
| W100BC2 | 137.21 ± 11.29 a | 306.15 ± 22.36 a | 98.12 ± 10.22 a | 541.48 ± 35.60 a |
| W100BC3 | 137.56 ± 12.69 a | 305.47 ± 18.97 a | 99.41 ± 13.88 a | 542.44 ± 39.55 a |

With a sample size of 3, all results are presented as the mean plus or minus the standard error. A statistically significant difference between treatments is indicated by different letters in the same column ($p < 0.05$).

**Table 3.** Modifications to the phosphorus content (g kg$^{-1}$) of *P. euphratica* seedlings exposed to varying water conditions in soil amended with varying concentrations of biochar.

| | Root P Concentration (g kg$^{-1}$) | Stem P Concentration (g kg$^{-1}$) | Leaf P Concentration (g kg$^{-1}$) | Total P Concentration (g kg$^{-1}$) |
|---|---|---|---|---|
| W30BC0 | 1.06 ± 0.11 b | 1.38 ± 0.17 b | 2.17 ± 0.33 b | 4.71 ± 0.38 b |
| W30BC1 | 1.15 ± 0.10 b | 1.44 ± 0.15 b | 2.26 ± 0.28 b | 4.85 ± 0.42 b |
| W30BC2 | 1.22 ± 0.11 b | 1.50 ± 0.18 a | 2.33 ± 0.25 b | 5.05 ± 0.41 b |
| W30BC3 | 1.18 ± 0.12 b | 1.45 ± 0.16 b | 2.28 ± 0.31 b | 4.91 ± 0.37 b |
| W60BC0 | 1.16 ± 0.11 b | 1.42 ± 0.14 b | 2.25 ± 0.30 b | 4.83 ± 0.35 b |
| W60BC1 | 1.26 ± 0.10 a | 1.56 ± 0.15 a | 2.38 ± 0.27 a | 5.20 ± 0.40 a |
| W60BC2 | 1.35 ± 0.13 a | 1.66 ± 0.19 a | 2.57 ± 0.32 a | 5.58 ± 0.39 a |
| W60BC3 | 1.28 ± 0.09 a | 1.58 ± 0.17 a | 2.48 ± 0.26 a | 5.07 ± 0.38 b |
| W100BC0 | 1.21 ± 0.11 b | 1.45 ± 0.16 b | 2.30 ± 0.31 a | 4.96 ± 0.37 b |
| W100BC1 | 1.21 ± 0.10 b | 1.51 ± 0.18 a | 2.33 ± 0.35 a | 5.05 ± 0.41 b |
| W100BC2 | 1.24 ± 0.12 a | 1.52 ± 0.15 a | 2.38 ± 0.29 a | 5.14 ± 0.42 b |
| W100BC3 | 1.23 ± 0.08 a | 1.50 ± 0.15 a | 2.37 ± 0.24 a | 5.10 ± 0.40 b |

With a sample size of 3, all results are presented as the mean plus or minus the standard error. A statistically significant difference between treatments is indicated by different letters in the same column ($p < 0.05$).

### 3.2. Soil Phosphorus Availability and Forms

Soil phosphorus availability and forms were altered by both biochar and water (Figure 1). The total phosphorus concentration varied between 0.34 and 0.44 g kg$^{-1}$, with ranges for Olsen-P, MBP, AL-P, Fe-P, Ca-P, and other elements ranging from 10.96 to 23.27 mg kg$^{-1}$, 13.01 to 25.09 mg kg$^{-1}$, 1.92 to 14.33 mg kg$^{-1}$, and 13.68 to 24.82 mg kg$^{-1}$, respectively. Compared with the soil without biochar addition under the W30 water condition, Olsen-P, MBP, and Ca-P were significantly increased by 28%, 14%, and 12% under the W60 water condition. The W100 water condition significantly increased Olsen-P by 16%. Olsen-P rose initially and subsequently declined with an increase in water content in the remaining soils with varying amounts of biochar added. In contrast, AL-P, Fe-P, Total-P, and MBP increased considerably with an increase in water content. Compared with the soil without biochar, Olsen-P, MBP, Ca-P, and Total-P increased by 35%, 35%, 24%, and 22%, respectively, under the condition of W60 under the treatment of biochar addition in BC2. The highest content of AL-P and Fe-P was in the treatment of the BC3 biochar addition under the W100 water condition. The highest Ca-P content was in the BC2 biochar addition treatment under the W30 water condition. The highest content of Total-P and MBP was in the BC2 biochar addition treatment under the W100 water condition. The highest Olsen-P content was in the BC2 biochar addition treatment under the W60 water condition.

### 3.3. Soil Phosphorus Cycle-Related Enzymes

Both biochar and water significantly changed the enzyme activities related to the phosphorus cycle in the rhizosphere soil of *P. euphratica* seedlings (Figure 2). In comparison to the soil without biochar under W30 water conditions, the activities of pyrophosphatase, alkaline phosphatase, acid phosphatase, and phytase were considerably elevated by 8%, 18%, 13%, and 34%, respectively, under W60 water conditions. Under W100 water conditions, the activities of alkaline phosphatase and neutral phosphatase were considerably boosted by 37% and 15%, respectively. The results showed that in the other soils that had different amounts of biochar, alkaline phosphatase rose sharply as water content rose, acid phosphatase fell sharply, and pyrophosphatase, phosphodiesterase, phytase, and neutral phosphatase rose sharply before falling. Compared with the soil without biochar, alkaline phosphatase, phytase, and neutral phosphatase increased by 23%, 28%, and 8%; 48%, 33%, and 47%; and 52%, 65%, and 16% under the conditions of W30, W60, and W100, respectively, under the biochar addition treatment of BC1. The highest alkaline phosphatase activity was the BC1 biochar addition treatment under the W100 water condition, and the strongest phosphodiesterase, pyrophosphatase active acid phosphatase, phytase, and neutral phosphatase were the BC1, BC0, BC3, and BC1 biochar addition treatments under the W60 water condition, respectively.

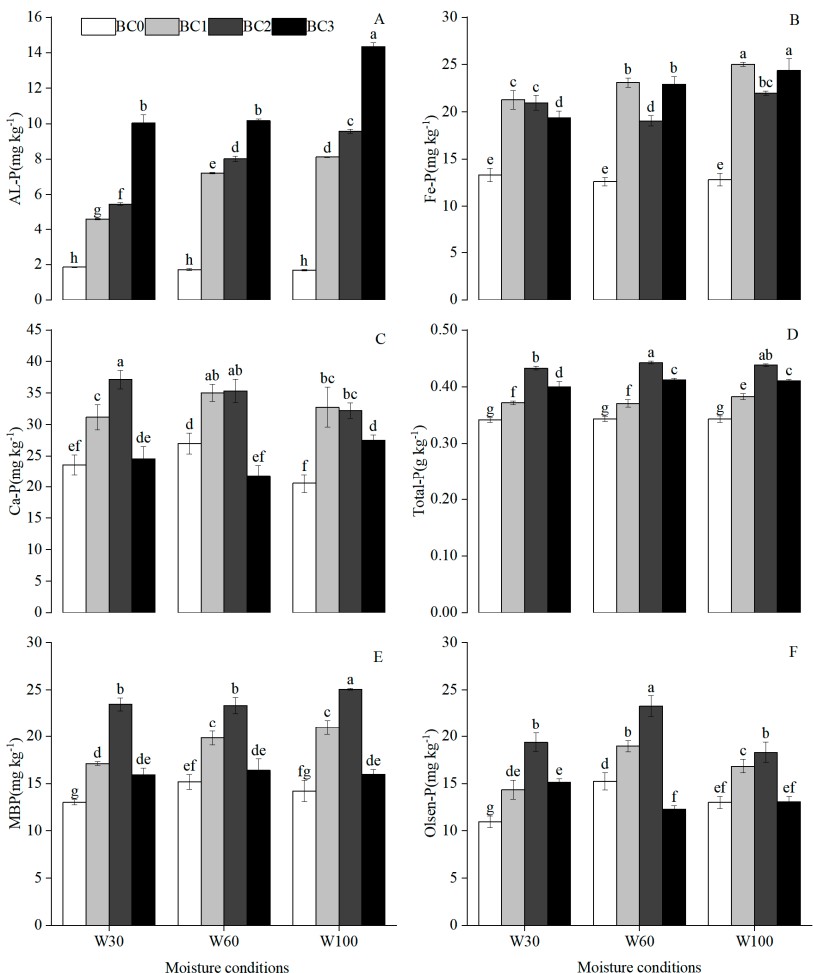

**Figure 1.** Figure (**A–F**) shows the effects of different biochar concentrations and water treatment on the contents of AL-P, Fe-P, Ca-P, Total-P, MBP and Olsen-P in rhizosphere soil of Populus euphratica seedlings, respectively. The range of potential values for the mean is shown by the error bar with a sample size of 3. The column displaying different lowercase letters indicates that there were notable variations in treatments ($p < 0.05$).

*3.4. Soil Microbial Community Structure*

The relative abundance of the microbial dominant genus (top 20) in each treatment is shown in Figure 3. Compared to other treatments, W30BC2, W60BC2, and W100BC2 had a much greater relative abundance of MND1 and RB41. *Sphingomonas's* relative abundance was much greater in the W60BC3 and W100BC3 treatments compared to the other treatments. When comparing treatments with and without biochar addition, we find that *Nitrospira*, *Steroidobacter*, *Ilumatobacter*, *Haliangium*, *Subgroup 10*, *P3OB 42*, and *Chryseolinea* are much more abundant in the former. This is true regardless of the water conditions. *Lysobacer* had the greatest relative abundance under W60 in the biochar-free treatment that tested various water conditions. *Sphingomonas's* relative abundance was greatest in the W30 treatment, which included adding BC1 biochar to various water conditions. The b-diagram of Figure 3 shows the fluctuation of the abundance of each fungal genus. *Fusarium* was found to have a much higher relative abundance in the W30BC3, W60BC3, and W100BC3 treatments compared to the others. The relative abundance of *Myriococcum* was noticeably increased in the W30BC1, W60BC1, and W100BC1 treatments compared to the other treatments. *Fusarium* and *Myriococcum* were found to have a much-increased relative abundance in treatments that included biochar under various water conditions compared to treatments that did not. W60 had the greatest relative abundance of *Botryotrichum* in the biochar-free treatment that tested various water conditions. In the

treatment of BC3 biochar addition under varying water conditions, the relative abundance of *Fusarium* was highest under W30.

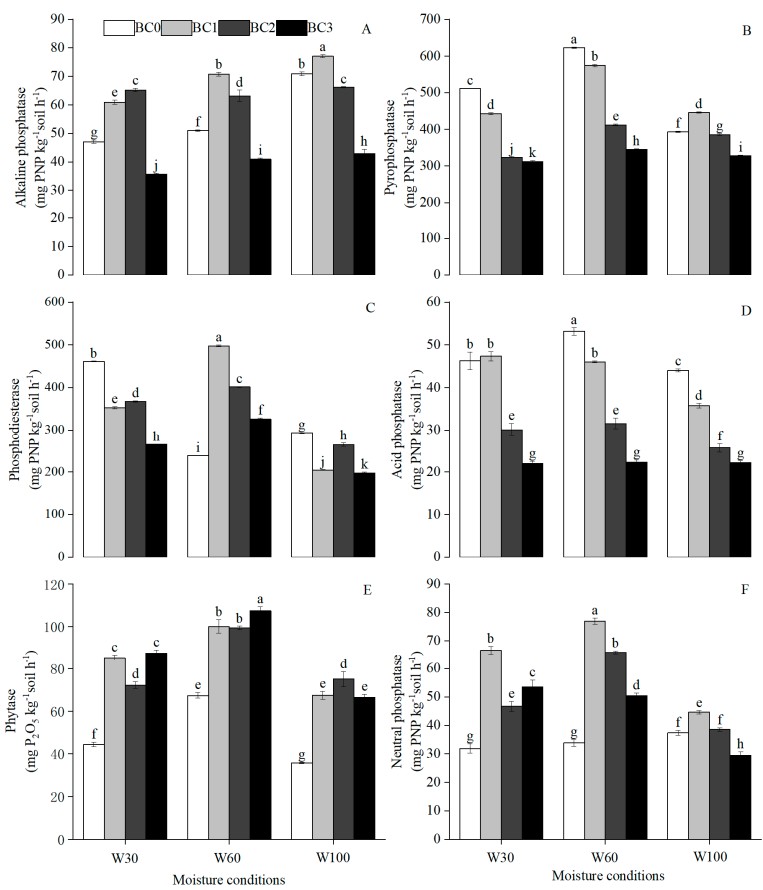

**Figure 2.** Figure (**A–F**) shows the effects of different biochar concentrations and water treatment on the activities of alkaline phosphatase, pyrophosphatase, phosphodiesterase, acid phosphatase, phytase and neutral phosphatase in rhizosphere soil of *P. euphratica* seedlings, respectively. With a sample size of 3, the error bar shows the range of possible values for the mean. There were significant differences between treatments ($p < 0.05$) shown by different lowercase letters on the column.

### 3.5. Functional Genes Related to Soil Phosphorus Cycle

According to Figure 4, the number of functional genes associated with the phosphorus cycle in the rhizosphere soil of *P. euphratica* seedlings was significantly affected by biochar. No significant influence was seen on *gcd*, *phoC*, *phoD*, and *pqqc* copy numbers under the W60 and W100 water circumstances compared to the soil without biochar under the W30 water condition. Biochar treatment considerably raised the copy counts of *gcd*, *phoC*, *phoD*, and *pqqc* in comparison to no treatment. When compared to the control group that did not receive biochar, the number of *gcd* copies increased by 57% when treated with W60 + BC3. As compared to the non-biochar treatment control group, those who did receive W60 + BC2 had increases of 78%, 39%, and 53% in the copy counts of *phoC*, *phoD*, and *pqqc*, respectively.

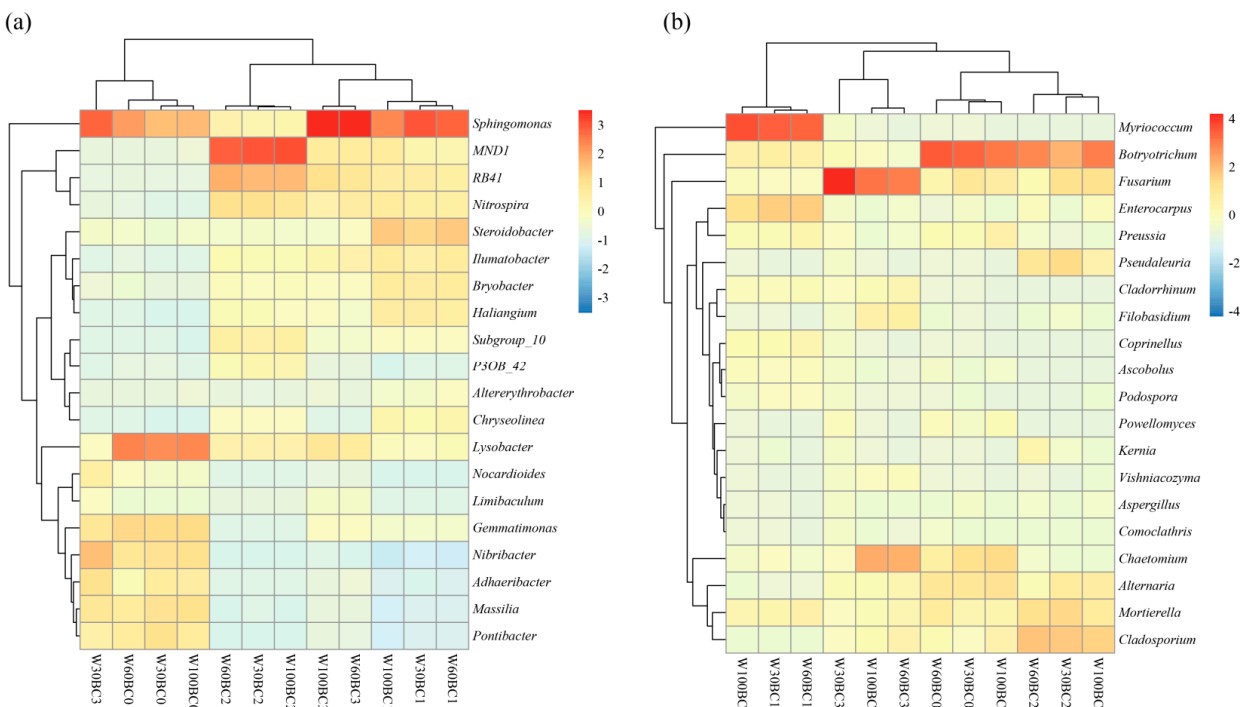

**Figure 3.** The abundance of bacteria and fungi in the first 20 OTUs in the rhizosphere soil of *P. euphratica* seedlings treated with varying quantities of biochar in various water conditions was analyzed using hierarchical clustering and a heat map, figure (**a**) for bacteria and figure (**b**) for fungi. In the heat map, each sample is represented by a column, and each categorization level is represented by a row. The standard deviation of the mean value, which represents the abundance of gene species, is color-coded. As a general rule, red denotes abundance while blue denotes scarcity.

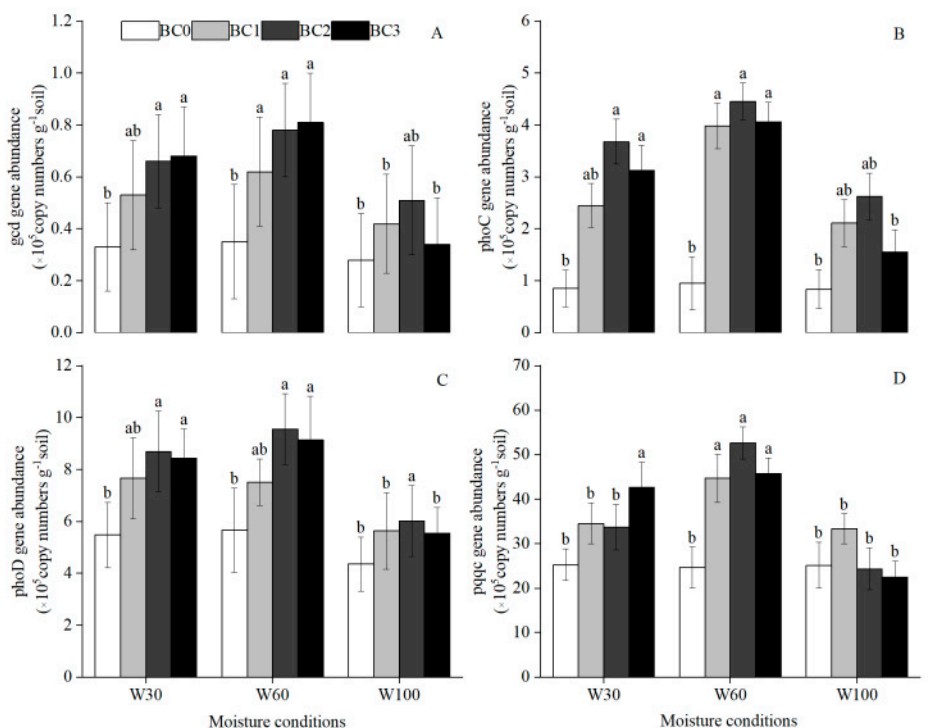

**Figure 4.** Figure (**A–D**) represents the changes of copy numbers of phosphate mineralization and solubility enhancing genes gcd, phoC, phoD and pqqc in rhizosphere soil of populus euonymus

treated with different biochars under different water conditions, respectively.When there are three observations, the error bar shows the margin of error around the mean. The presence of different lowercase letters in the column denotes significant variations among treatments ($p < 0.05$).

Network analysis based on Spearman correlation analysis was conducted to clarify the possible co-occurrence patterns of microbial phyla (fungi and bacteria) and phosphorus transformation functional genes (*gcd*, *phoC*, *phoD*, and *pqqc*). At the 0.05 level, there is a significant change. The functional genes for phosphorus conversion are shown in Figure 5, along with fifteen bacterial phyla and fourteen fungal phyla. A set of interconnected genes is called a module, and the genes with the greatest number of connections are called hubs. There are 33 gates or genes in the biggest module.

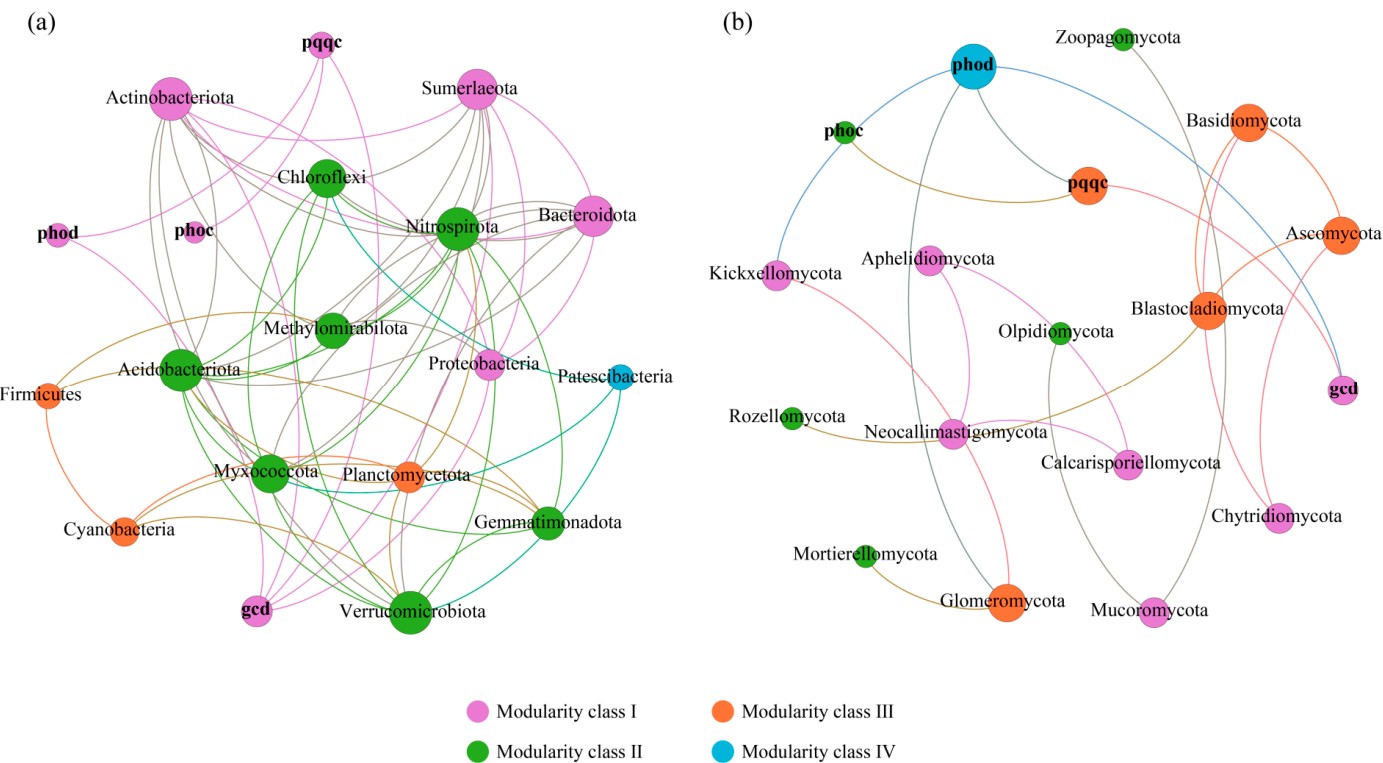

**Figure 5.** The phylum of microbes, the frequency of their co-occurrence, and the genes involved in phosphorus transformation (*gcd*, *phoC*, *phoD*, and *pqqc*) were analyzed using a network approach, figure (**a**) for bacteria and figure (**b**) for fungi.

## 4. Discussion

### 4.1. Influence of Soil Characteristics on P. euphratica Seedling Development

Table 1 shows that many soil parameters including pH, CEC, SOM, total carbon, total nitrogen, total potassium, total aluminum, total iron, and AN and AK concentrations increased as biochar application increased, all under the same water content. The biochar ash preserved a greater concentration of base ions, including $k^+$, $Ca^{2+}$, and $Na^+$, which could be a result of the pyrolysis carbonization [26]. Biochar made from cow dung had a pH of 10.14 when utilized in this study. Soil pH rose in direct correlation with the biochar addition fraction. Water significantly impacted crop development in this research. Crop development was much enhanced with more water, and biochar affected *P. euphratica* seedlings in a major way as well (Table 2). The growth is best when BC2 biochar is added, and the growth of crops is inhibited when the amount is high, which is similar to the conclusion of Marks et al.'s study on the short-term growth of ryegrass using poplar biochar on calcareous soil [27]. This result may be related to the alkalinity of the basic soil, because a large number of reports on biochar promoting crop growth have been studied on acidic soils [28]. It may also be due to the high salt content of biochar, which reduces

the soil solute potential [29]. The lower the soil water potential, the lower the soil water availability, indicating that under the same water conditions, the greater the amount of biochar added, the more difficult it is for crops to absorb water [30]. Phosphorus was most concentrated in the roots, stems, and leaves of *P. euphratica* seedlings when the water was at W60, and it was inhibited when the addition was high (Table 3). It shows that biochar limits the absorption of phosphorus, and a high addition of biochar slightly inhibits the concentration of phosphorus, which may be due to the toxic excitatory effect of organisms in toxicology [31].

### 4.2. Effects of Soil Phosphorus Availability and Forms

Under the same water circumstances, soils that had biochar added had much higher total phosphorus and Olsen-P contents compared to those that did not (Figure 1D,F). According to this research, biochar has a far higher total phosphorus content than soil. Thus, biochar's use has the potential to boost soil total phosphorus concentration [32]. There are three main explanations for why biochar increases the soil Olsen-P content. To begin with, there is some phosphorus in biochar. The carbonization process releases phosphate from the biochar source material, which is cow dung. Simultaneously, the biochar retains nearly all of the phosphorus present in the cow dung used as a source material. The incorporation of biochar into soil transforms this phosphorus into a soil phosphorus source [33]. Secondly, the chemical composition of the soil may have been altered by biochar, which might indirectly impact the availability and form of phosphorus in soils that are naturally alkaline. Figure 1A,B show that under the same water circumstances, soil with biochar added had significantly higher levels of Al-P and Fe-P compared to soil without. Through its capacity to adsorb ions in soil that are readily reactable with phosphorus and precipitate, biochar mitigated the formation of metal–phosphorus complexes, like Al-P and Fe-P, caused by soluble phosphorus and the $AL^{3+}$ and $Fe^{3+}$ ions released during the high-temperature thermal decomposition of mixed raw materials. When organic molecules or substances adsorbed on the biochar's surface form chelates and release sequestered phosphorus, it improves soil phosphorus availability. Figure 3 shows that soil microorganism biomass, including phosphate-solubilizing bacteria and fungi like *Sphingomonas* and *Fusarium*, was enhanced by charcoal addition and water conditions. Under the same water conditions, the soil MBP content in this investigation was also noticeably greater when compared to the treatment without biochar (Figure 1E). These phosphate-solubilizing microorganisms release phosphatases and organic acids, which can transform some of the organic phosphorus in the soil into usable phosphorus [34]. One such theory is that the soil's insoluble AL-P, Fe-P, and Ca-P were activated when charcoal and water were added. A possible explanation for the observed positive stimulation impact of biochar treatment on phosphorus fixation is that, in this investigation, AL-P content increased with increasing biochar content (Figure 1A) under the same water conditions. Soil iron and phosphorus concentrations declined when biochar concentrations rose, according to our research (Figure 1B).

### 4.3. Effects of Soil Phosphorus Cycle-Related Enzymes

Soil phosphatase activity is a key measure of phosphorus supply capacity, alongside Olsen-P. In contrast to the observed trend in soil pH value changes, this study found that soil acid phosphatase reduced as charcoal addition increased under the same water treatment conditions (Figure 2D). The reason behind this is that biochar effectively neutralized soil acidity and decreased soil acid phosphatase concentration. Results were comparable for Zhang et al. as well. They discovered that the right amount of biochar treatment could greatly boost soil acid phosphatase levels, but that excessive biochar treatment somewhat reduced acid phosphatase activity; they hypothesized that this could be because biochar's alkaline properties raise soil pH [35]. The content of soil alkaline phosphatase mainly showed a trend of increasing first and then decreasing with an increase in biochar addition (Figure 2A). This may be the result of the combined action of water and biochar. On the

one hand, flooding can significantly reduce soil alkaline phosphatase. On the other hand, biochar has a certain degree of the excitation effect, which is consistent with the results of Khadem's research [36]. The fact that the magnesium and zinc ions found in biochar itself can be utilized as coenzyme components of alkaline phosphatase may be the primary cause of the increase in soil alkaline phosphatase activity. Additionally, the original stable trace elements in soil can be activated by biochar at the same time. Therefore, biochar can affect soil alkaline phosphatase activity by increasing the coenzyme factors of phosphatase in soil [37]. It is possible that biochar-induced alterations in phosphorus fractions are responsible for the observed changes in phosphomonoesterase and phosphodiesterase activity in this investigation. According to Figure 2C, biochar can reduce the rate of phosphodiesterase activity while increasing alkaline phosphatase activity, suggesting that biochar can delay the mineralization of phosphate diester to phosphate monoester but speed up the hydrolysis of phosphate monoester to direct phosphate.

*4.4. Effects of Soil Microorganisms and Phosphorus Transformation Functional Genes*

Cow dung biochar changed the amount and composition of soil microbes, according to the study's results. More specifically, phosphate-solubilizing bacteria and fungi, including *Fusarium* and *Sphingomonas*, were found in higher quantities (Figure 3). This is consistent with previous studies [38]. Microorganisms find it difficult to use biochar as a carbon source because of its high stability. The rise in its amount and activity could be attributed to the fact that biochar's porous nature allows it to absorb fertilizer and water, creating an environment that protects microorganisms from predators [39]. The total ratio of bacteria to fungi, dominant species, and community composition have changed [40]. It should be mentioned that biochar can serve as a home for bacteria with pore sizes ranging from 0.3 to 3.0 µm, but it might not be large enough to support the colonization and growth of fungi with pore sizes ranging from 2 to 80 µm, according to this study [41]. Fungi may not be able to penetrate the tiny holes in biochar, which might be bad for their growth, because bacteria can absorb water and nutrients from these holes [42]. For example, Aspergillus phosphate-solubilizing fungi did not change significantly under different biochar treatments (Figure 3b). In addition, in this study, phosphorus-solubilizing bacteria such as *Gemmatimonas* and *Massilia* were reduced under high biochar addition treatments, which may be due to the fact that toxic compounds carried and adsorbed by biochar may inhibit microbial activity and thus affect soil function [43].

It has been discovered that the amount of biochar added to soil affects microbial reproduction [44]. In our investigation, there was a high link between changes in the number of P-transforming genes and the abundance of bacterial and fungal communities (Figure 5). Studies have indicated that the addition of biochar to soil can have a significant impact on the variety and arrangement of microorganisms related to the phosphorus cycle [45]. One possible explanation for the shift in gene abundance is the effect of biochar on phosphate-solubilizing microbes through changes to soil properties. Soil biochar has the potential to alter the phosphate-solubilizing bacterial number and activity by influencing soil pH, CEC, C/P ratio, and phosphorus availability, among other soil parameters [46]. The study found a correlation between the relative abundance of certain bacterial and fungal phyla and the gene copy numbers of *phod*, *phoc*, *pqqc*, and *gcd*. The abundance of *Actinobacteriota*, *Sumerlaeota*, *Bacteroidota*, and *Proteobacteria* is correlated with that of *phod*, *phoc*, *pqqc*, and *gcd*, whereas the abundance of *Basidiomycota*, *Blastocladiomycota*, *Ascomycota*, and *Glomeromycota* are correlated with that of *pqqc*. Several phyla of microbes share a common ancestor with the phosphorus transformation functional genes (*gcd*, *phoC*, *phoD*, and *pqqc*), which are involved in determining the make-up of bacterial and fungal communities. Soil phosphorus availability can be impacted by several factors, which can impact various soil phosphorus transformations and potentially serve as markers for assessing their occurrence.

## 5. Conclusions

The availability of phosphorus in the soil, the composition of the microbial community, and the expression of functional genes in microorganisms were some of the characteristics of the soil that were demonstrated to be impacted differentially by biochar depending on the water conditions. Biochar is a potent approach that can drastically alter soil pH, CEC, total C, and SOM. Additionally, it can enhance the phosphorus concentration of microbial biomass and soil inorganic phosphorus components. Biochar is a powerful technology that provides a return on investment. The reintroduction of biochar into the soil resulted in changes to the composition of bacterial and fungal communities in the rhizosphere soil. These changes included an increase in the copy counts of soil phosphorus functional genes such as *phoC*, *gcd*, and *pqqc*. Another effect of the biochar treatment was an alteration in the number of bacteria that were able to dissolve phosphate. In conclusion, carbonizing cow dung and then reintroducing it to the soil could be an environmentally acceptable method of boosting the availability of phosphorus in the soil, which in turn can be beneficial to the phosphorous health of crops. In order to improve plantation management, there should be a reduction in the use of chemical fertilizers and an increase in the utilization of biochar. It is vital to examine and analyze the methods in which cow dung biochar and *P. euphratica* seedlings are utilized over an extended period of time in order to uncover the most scientific method of incorporating these two components.

**Supplementary Materials:** The following supporting information can be downloaded at: https://www.mdpi.com/article/10.3390/f15050831/s1, Table S1, Data accession information.

**Author Contributions:** Conceptualization, Y.F.; methodology, Y.F.; formal analysis, Y.F. and Y.C.; resources, G.L., Y.F. and Y.C.; data curation, G.L., Y.F. and Y.C.; writing—review and editing, Y.F.; supervision, G.L.; project administration and funding acquisition, G.L. All authors have read and agreed to the published version of the manuscript.

**Funding:** This research was funded by National Natural Science Foundation of China (42171026) and National Natural Science Regional Foundation (32260266).

**Data Availability Statement:** The data presented in this study are available on request from the corresponding author.

**Conflicts of Interest:** The authors declare that they have no known competing financial interests or personal relationships that could have appeared to influence the work reported in this paper.

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
