# Peer review of "Effects of Biochar on Soil Inorganic Phosphorus Components, Available Phosphorus, Enzyme Activities Related to Phosphorus Cycle, Microbial Functional Genes, and Seedling Growth of Populus euphratica under Different Water Conditions"

_forests, doi:10.3390/f15050831_

Round 1

Reviewer 1 Report

Comments and Suggestions for Authors

The article presents a significant number of experiments and interesting results regarding the application of biochar derived from cow manure.

The authors must correct some information, as indicated below:

 Micromonas does not belong to Actinomycetes; it is a genus of green picoplankton (Chlorophyta).

Specify the primers used for bacterial PCR, both forward and reverse, e.g., 338F, and correct the terminal ends (5' and 3').

Provide the same detail for fungal PCR, specifying the upstream primer.

In the results, the diversity of microorganisms obtained is at genus level, not species level.

Review the use of "charcoal" as it is not synonymous with "biochar"; generally, reference is not made to "charcoal biochar."

I suggest that in the titles of the tables, only the descriptor remains, and detailed information is moved to the table footer.

Several ortho-typographic errors need to be addressed to polish the manuscript:

  To adhere to the rules of citing the species in question, the first time it is mentioned, write it out completely as "Populus euphratica," and subsequent times it is not necessary to repeat the genus; abbreviate it as "P. euphatica."

 Also, in both cases, they should be italicized, which is not consistent throughout the text. The same goes for several of the genera mentioned (e.g., Penicillium, Bacillus, Fusarium).

 Additionally, genes should be italicized.

 Correct the superscripts for units such as cm^3, m^2, hm^2, as well as P2O5, and the superscripts for ions K+, Ca2+, and Na+.

 Replace "microliters" with "µL."

 There are many missing spaces after the period at the end of sentences.

It is more common to refer to Control than to classify it as BC0 or its default is to specify it as such to confuse the reader as if it were another treatment.

Generally significant differences are usually indicated with *

Author Response

Response to Reviewer# 1

Response: We have carefully revised the manuscript based on the reviewers' suggestions.The changes we have made are presented with revision mode in the marked revised manuscript.We would be very grateful if the manuscript could be published in Forests.The following part is the point-by-point responses to the reviewer# 1.

1.Micromonas does not belong to Actinomycetes; it is a genus of green picoplankton (Chlorophyta).

=> Response: Thank you for pointing out the error, we have deleted this error in line 108 of the manuscript.

2.Specify the primers used for bacterial PCR, both forward and reverse, e.g., 338F, and correct the terminal ends (5' and 3').

=> Response:Thank you for your kind reminder, we have perfected the information of bacterial primers in line 253 of the manuscript.

3.Provide the same detail for fungal PCR, specifying the upstream primer.

=> Response:Thank you for your kind reminder, we have supplemented the information of fungal primers in line 257 of the manuscript.

4.In the results, the diversity of microorganisms obtained is at genus level, not species level.

=> Response:Thank you for pointing out the error, we have corrected the information described in the microbial diversity results in line 396 of the manuscript.

5.Review the use of "charcoal" as it is not synonymous with "biochar"; generally, reference is not made to "charcoal biochar."

=> Response:Thank you for pointing out the error. We have deleted this error in line 470 of the manuscript.

6.I suggest that in the titles of the tables, only the descriptor remains, and detailed information is moved to the table footer.

=> Response:Thank you for your pertinent suggestions. We have revised the table titles in lines 330, 334 and 340 of the manuscript.

  1. To adhere to the rules of citing the species in question, the first time it is mentioned, write it out completely as "Populus euphratica," and subsequent times it is not necessary to repeat the genus; abbreviate it as " euphatica."

=> Response:Thank you for your pertinent suggestions, we have revised the abbreviation of ' Populus euphratica ' in the manuscript.

  1. Also, in both cases, they should be italicized, which is not consistent throughout the text. The same goes for several of the genera mentioned (e.g., Penicillium, Bacillus, Fusarium).

=> Response:Thanks to your careful corrections, we have made italic corrections to microbial names such as Penicillium, Bacillus, and Fusarium in the manuscript.

9.Additionally, genes should be italicized.

=> Response:Thank you for your careful correction. We have made italic corrections to gene names in the manuscript.

10.Correct the superscripts for units such as cm^3, m^2, hm^2, as well as P2O5, and the superscripts for ions K+, Ca2+, and Na+.

=> Response:Thank you for your careful correction. We have correctly marked the units of P2O5, m ^ 2, m ^ 2, hm ^ 2, K + and Ca2 + in the 55th, 172nd and 177th lines of the manuscript.

11.Replace "microliters" with "µL."

=> Response:Thanks for your pertinent advice, we have replaced the ' microliters ' in the manuscript with ' μL '.

12.There are many missing spaces after the period at the end of sentences.

=> Response:Thank you for your pertinent suggestion, we have deleted the extra blank at the end of the sentence in the manuscript.

13.It is more common to refer to Control than to classify it as BC0 or its default is to specify it as such to confuse the reader as if it were another treatment.

=> Response:Thank you for your pertinent suggestion. We have explained it in line 192 of the manuscript.

14.Generally significant differences are usually indicated with *

=> Response:Thanks for your sincere reminder, we have marked * in the manuscript.

Reviewer 2 Report

Comments and Suggestions for Authors

Comments and Suggestions for Authors

Title: Effects of biochar on soil phosphorus availability, phosphatase activity, microbial functional genes and growth of Populus euphratica seedlings under different water conditions

Dear Authors

 The subject corresponding to the Forests journal’s profile. The manuscript presents the impact of biochar on the content of phosphorus forms in the soil, the activity of selected enzymes, microbial functional genes and growth of Populus euphratica seedlings under different water conditions. The manuscript presents many interesting research results. I think that the results obtained in one year of research may not be sufficient to draw correct conclusions. However, many corrections and additions need to be made.

Remarks

In order to increase the usefulness of the article, Authors must refer to the following points. Additions should be made to increase the scientific value of the manuscript.

1.       The title of the manuscript should be corrected. After all, various forms of phosphorus in the soil and enzymatic activity, not only phosphatase, were presented.

2.       Abstract: The purpose of research needs to be improved. Units should be presented according to editorial requirements.

3.       Introduction: Much of this section is loosely related to the results presented in the manuscript. So I recommend shortening this part. I believe this may help reduce plagiarism repetitions. Line 58 - phosphorus is not a mineral. The Latin generic names of soil microorganisms should be written in italics.

4.       Materials and Methods: Subsection 2.1. -The soil type should be provided according to the WRB 2022 classification. Subsection 2.2. Lines 179, 183-185 Please write the units correctly (see subsection 2.1.). Subsection 2.3. The principle of the method for determining phosphorus in plant parts (reaction reagents, apparatus) should be described. Line 234 - not included in References. Subsection 2.5 should describe the determination of phosphorus forms in soil, which is presented in subsection 3.2. A subsection should be added regarding the principles of determining enzymatic activity in soil.

5.       Results: The tables need to be improved technically (font, size). Table 1 - total C should be given in g/kg, the same as SOM. Subsection 3.2. Ga-P what phosphorus is it? or maybe Ca-P? Please provide the full name of the MBP abbreviation. The descriptions of Figures 1-5 should be corrected. Please follow the order (e.g. Figures 1A to 1F). Figures 1D - the total P content in the soil should be given in g.kg-1.

6.       Discussion: Line 454 - It should be: K+, Ca2+.... Line 466 - [32] should be added. The Discussion section should be shortened. Subsection 4.2 - citations should be corrected according to subsequent References numbers. In subsection 4.2, please add references to pH, which determines the content of available forms of phosphorus in the soil to the greatest extent.

7.       Please add an Authors Contributions note after the Conclusions section.

8.       References: Should be corrected in accordance with publishing requirements.

Best regards

Author Response

Response to Reviewer# 2

Response: We have carefully revised the manuscript based on the reviewers' suggestions.The changes we have made are presented with revision mode in the marked revised manuscript.We would be very grateful if the manuscript could be published in Forests.The following part is the point-by-point responses to the reviewer# 2.

1.The title of the manuscript should be corrected. After all, various forms of phosphorus in the soil and enzymatic activity, not only phosphatase, were presented.

=> Response: Thanks for your pertinent suggestion, we have changed the title of the manuscript to“Effects of biochar on soil inorganic phosphorus components, available phosphorus, enzyme activities related to phosphorus cycle, microbial functional genes and seedling growth of Populus euphratica under different water conditions”.

2.Abstract: The purpose of research needs to be improved. Units should be presented according to editorial requirements.

=> Response: Thanks for your careful reminding, we have upgraded the research purpose in line 12 of the article. And we have regulated the use of units.

  1. Introduction: Much of this section is loosely related to the results presented in the manuscript. So I recommend shortening this part. I believe this may help reduce plagiarism repetitions. Line 58 - phosphorus is not a mineral. The Latin generic names of soil microorganisms should be written in italics.

=> Response: Thanks to your wise advice, we have made some cuts in the introduction of the manuscript. And the part about phosphorus being a mineral was removed. We have also changed the Latin names of soil microbes to italics.

 4.Materials and Methods: Subsection 2.1. -The soil type should be provided according to the WRB 2022 classification. Subsection 2.2. Lines 179, 183-185 Please write the units correctly (see subsection 2.1.). Subsection 2.3. The principle of the method for determining phosphorus in plant parts (reaction reagents, apparatus) should be described. Line 234 - not included in References. Subsection 2.5 should describe the determination of phosphorus forms in soil, which is presented in subsection 3.2. A subsection should be added regarding the principles of determining enzymatic activity in soil.

=> Response: Thank you for pointing out the error in the manuscript. We have deleted the description of the soil type and corrected the error in the unit. At the same time, we have added the method principle (reaction reagent, device) for the determination of the plant part in line 196 of the manuscript. We have deleted the experimental method in line 234 and redescribed it. In line 224 of the manuscript, we added the content on the determination method of soil phosphorus form, and in line 228, we added a section on the determination of enzyme activity related to soil phosphorus cycle.

5.Results: The tables need to be improved technically (font, size). Table 1 - total C should be given in g/kg, the same as SOM. Subsection 3.2. Ga-P what phosphorus is it? or maybe Ca-P? Please provide the full name of the MBP abbreviation. The descriptions of Figures 1-5 should be corrected. Please follow the order (e.g. Figures 1A to 1F). Figures 1D - the total P content in the soil should be given in g.kg-1.

=> Response: Thanks to your wise advice, we have optimized the table and the total carbon determination is determined using combustion infrared absorption spectrometry using a high frequency infrared carbon sulfur analyzer (CS844, LECO, USA), so the units are % used. We corrected the Ca-P in the manuscript and also corrected the Ca-P in the figure. We have also provided the full name of the MBP abbreviation in the manuscript, and the description in Figure 1 has been modified. The unit of total phosphorus in soil was also changed to g kg-1.

 6.Discussion: Line 454 - It should be: K+, Ca2+.... Line 466 - [32] should be added. The Discussion section should be shortened. Subsection 4.2 - citations should be corrected according to subsequent References numbers. In subsection 4.2, please add references to pH, which determines the content of available forms of phosphorus in the soil to the greatest extent.

=> Response: Thanks for your careful reminding, we have corrected the writing on K+Ca2+, deleted the discussion section, added a reference to the description of pH, and finally updated the reference number of the whole paper.

  1. Please add an Authors Contributions note after the Conclusions section.

=> Response: Thanks for your kind reminder, we have added the Authors Contributions note on line 610 of the manuscript.

8.References: Should be corrected in accordance with publishing requirements.

=> Response: Thanks for your kind reminder, we have updated the manuscript reference format in accordance with the journal requirements.

Round 2

Reviewer 2 Report

Comments and Suggestions for Authors

Comments  for Authors

Title: Effects of biochar on soil phosphorus availability, phosphatase activity, microbial functional genes and growth of Populus euphratica seedlings under different water conditions

Dear Authors

The manuscript (file Word forests-2972045-coverletter)  has been corrected and supplemented in accordance with the comments.

Best regards

Author Response

Response to Reviewer# 2(Round 2)

Response: We have carefully revised the manuscript based on the reviewers' suggestions.The changes we have made are presented with revision mode in the marked revised manuscript.We would be very grateful if the manuscript could be published in Forests.The following part is the point-by-point responses to the reviewer# 2.(Round 2)

1.There needs to be a space between P. and Polulus.

=> Response: Thanks for your pertinent suggestion, We have added Spaces between P. and Polulus in the manuscript.

2.Delete ", according to research" in the abstract.

=> Response: Thanks for your careful reminding, We have deleted “according to research" from the abstract.

3.A sentence of "A previous forest resource ..." is italicized.

=> Response: Thanks to your wise advice, We have restored the italicized sentence "A previous forest resource..." in line 41 of the manuscript.

4.Please delete the space between numbers and %.

=> Response: Thank you for pointing out the error in the manuscript. We have deleted the Spaces between numbers and % in the manuscript.

5.Is "(1)" necessary in the last paragraph of the introduction?

=> Response: Thanks to your wise advice, We have deleted "(1)" in line 144 of the manuscript.

6."degree Celcius" - use the symbol for consistency.

=> Response: Thanks for your careful reminding, We have unified the use of the symbol "degree Celcius" in the manuscript.

7.What is W30BC2?  I suggest writing W30 & BC2 treatment in Section 3.4.

=> Response: Thanks for your kind reminder, We have used W30&BC2 instead of W30BC2 in Part 3.4.

8.There are unnecessary spaces in (Table 2) in Section 4.1.

=> Response: Thanks for your kind reminder, We have removed unnecessary Spaces in line 471 of the manuscript.

9.please superscript -1 in the caption of Table 3.

=> Response: Thanks for your kind reminder, We have superscripted -1 in the caption of Table 3.
